# Evaluating the Environmental Impact of Anthropogenic Activities on Human Health: A Systematic Review

**Luigi Cofone** [1] , **Marise Sabato** [1,*] , **Enrico Di Rosa** [2] , **Chiara Colombo** [2] and **Lorenzo Paglione** [2,3]

1 Department of Public Health & Infectious Diseases, Sapienza University of Rome, 00185 Rome, Italy; luigi.cofone@uniroma1.it

2 Department of Prevention, ASL Roma 1, 00193 Rome, Italy; enrico.dirosa@aslroma1.it (E.D.R.); chiara.colombo@aslroma1.it (C.C.)

3 Department of Civil, Constructional and Environmental Engineering, Sapienza University of Rome, 00185 Rome, Italy

\* Correspondence: marise.sabato@uniroma1.it

**Abstract:** Due to major global urbanisation, a careful evaluation of plans (town planning and mobility) and projects (industrial and development) is required in order to measure their impact on health and environmental matrices. In Italy, Legislative Decree No 152/06 introduced two procedures: the EIA (Environmental Impact Assessment) and SIA (Strategic Impact Assessment). Their focus, however, does not consider human health. Recently, the Integrated Environmental and Health Impact Assessment (IEHIA) was introduced; this defines the parameters necessary to provide an EIA that includes human health as a factor. This systematic review was conducted, including both the population impacted by new facilities and the method used to define their impact. Our database search produced 724 articles, of which 33 were eligible. Studies included landfill plans, manufacturing industries, mobility policies, energy production, and the environmental health of an area. All studies show how an approach encompassing multiple parameters can analyse the impact of a new facility in a comprehensive manner. This review shows that the use of health-related environmental impact parameters is essential for the integration of a project into a community, and can allow a wider understanding of the possible impacts on human health, both direct and indirect.

**Keywords:** Environmental Impact Assessment; Health Impact Assessment; determinants of health

## 1. Introduction

Increasing globalisation, industrialisation, and the development of transport systems and the trade sector are having a massive impact on the world's ecosystem, and, consequently, on the health of the global population. The large rise in the negative impact of these large-scale anthropogenic projects is alarming as it causes irreversible damage to the ecological balance of the ecosystem. In order to mitigate the various negative effects, the One Health Model approach provides several tools that can be used in upcoming projects; such as the Environmental Impact Assessment (EIA), which can be used in the design phase, and the Strategic Environmental Assessment (SEA), which can be used in the territorial planning phase. These tools allow the evaluation of different aspects of the long-term effects of a project on the surrounding environment [1–3].

The EIA first emerged in the US in 1969, with its National Environmental Policy Act (NEPA). Between 1973 and 1974, Australia, Canada, and New Zealand followed the NEPA statement by implementing it in their legislation and administrative procedures. In Europe, it was introduced with the "Council Directive 85/337/EEC of 27 June 1985" and amended with Directive 2011/92/EU and Directive 2014/52/EU [4–6]. Directive 2014/52/EU also includes "population" and "human health". The Directive does not, however, specify parameters on how health effects can be assessed. The EIA is widely used in many countries around the world today. As of 2017, most of these countries implemented

the EIA in their national legislations in various sectors, such as agriculture, tourism, mining, and so forth. Various parameters are measured, including soil, water, and air pollution, waste production, noise pollution, and loss of biodiversity.

In 2016, the World Health Organisation (WHO) tried to estimate environmental exposures and the effects of these on the burden of diseases afflicting humanity around the world. It has been found that the environment contributed up to 24% of deaths globally [7]. This shows how one of the most important social determinants of health, from an "equitable" point of view, is the physical environment in which one lives [8]. The living environment can by itself determine the probability of contracting or developing a disease [9]. Many studies have shown that low-income countries are more susceptible to climate change relative to high-income countries. Low-income countries are primarily affected with reference to health, food, water, and ambient pollution [10].

The Netherlands Environmental Assessment Commission and the United Nations Environment Programme (UNEP) state that the EIA is considered a legal requirement in 186 countries of the 193 nations recognised by the UN. The SEA, on the other hand, introduced in the EU in 2001 with Directive 2001/42/EC, is used in only about 60 UN countries [11,12]. With regard to Italy, the EIA was introduced in 1988, as the "*Valutazione Impatto Ambientale (VIA)*" (Environmental Impact Assessment), and is used mainly for projects and production facilities [13]. The SEA ("*Valutazione Ambientale Strategica (VAS)*") was introduced in 2008, and is now used in mobility and urbanisation planning. Today, the Italian Legislative Decree 152/06 is used nationally, though it is not considered binding and is used instead as a guideline. This Decree tends to have a very specific flaw as it focuses on the environment and, therefore, the health impact is not easily measured [14]. In Italy, the term "*Valutazione Impatto Sanitario*" (VIS) (Health Impact Assessment) is not yet mandatory, although there is a great push in trying to enhance the role of the Health Impact Assessment within Environmental Impact Assessments at a national level [15]. Project VIIAS, "*Valutazione Integrata dell'Impatto Ambientale e Sanitario*" (Integrated Environmental and Health Impact Assessment—IEHIA), a national project financed by the Ministry of Health, was recently introduced in Italy [16]. The IEHIA is analogous to the Health Impact Assessment (HIA), used internationally for the impact assessment of policies, plans, and projects. This work resulted from a desire to define the parameters necessary to provide an EIA that thoroughly includes human health, promoting a One Health Model approach, addressing the needs of the most vulnerable populations, animals, and the environment.

Unfortunately, the EIA and SEA cannot qualify for a full assessment of the environmental impact as they have a very limited scope in measuring the health impact on humans. Therefore, there have been different attempts to introduce the HIA in order to have a much more comprehensive view [17]. The HIA can be used in policy making as it uses an evidence-based medicine approach. The WHO, with its 1999 Consensus Paper, outlined methods that could be used to implement the HIA at an international level [18]. Nowadays, the HIA method is used in many countries, usually on a voluntary basis, as it is not incorporated in national legislation. In spite of this, it is now widely disseminated in many countries with the objective to reach the Sustainable Development Goals (SDGs), introduced by the UN in order to reach "equity" by 2030 [19]. Gulis et al. (2022) describes how the inclusion of SDGs indicators in EIA, SIA, and HIA are important in reaching the target of a much more sustainable world [20,21]. Since 2022, the One Health Model approach has been the next step in tackling the different threats caused by climate change and environmental impacts on health [22]. Again, the Health Impact is not easily estimated as many parameters are not considered. There is a wide range of tools available to be integrated into a single, more complete tool. Therefore, this means that a much more comprehensive and inclusive approach is needed from a public health perspective.

## 2. Materials and Methods

### 2.1. Selection Protocol and Search Strategy

The current systematic review was carried out according to the Preferred Reporting Items for Systematic Reviews and Meta-Analyses (PRISMA) Methodology [23]. The associated protocol was registered in PROSPERO with the following ID: CRD42024509337.

Research in the literature was performed using three different databases: PubMed, Scopus, and Web of Science. All articles were searched, from the beginning to the 5 February 2024 using the following search string: ("environmental impact assessment" OR "strategic environmental assessment") AND ("population health" OR "public health") AND (indicators OR method OR guidelines).

### 2.2. Inclusion Criteria for the Study

All articles found were first screened by title and abstract and secondly by the full text. Screening was conducted independently by all authors (L.C., M.S., E.D.R., C.C., L.P.). The same authors (L.C., M.S., E.D.R., C.C., L.P.) read the full texts independently. All doubts and disagreements were then discussed, and disagreements were solved by reaching consensus between the authors.

Any research that provided information regarding a specific example of a framework and an assessment of its influence on human health was considered acceptable. Reviews, meta-analyses, case studies, symposia, editorials, and other kinds of studies were not accepted. Only those articles presenting original data were admissible. All references of the included articles were examined; this was performed in order to identify the articles that were included in those references. Only articles published in Italian or English were included.

### 2.3. Data Extraction and Quality Assessment

Information regarding author, year, country, methods, and parameters used were collected from all studies. Furthermore, the data were organised to define the parameters that were important for the assessment of the environmental impact on human health.

A quality assessment was conducted by the use of the Newcastle–Ottawa Quality Assessment Scale (NOS).

## 3. Results

A total of 724 studies were retrieved from the following databases: PubMed, Web of Science, and Scopus. Of these, 115 duplicates were removed and 609 were screened according to title and abstract. In this step, 472 articles were excluded as they did not meet the inclusion criteria chosen by all the authors (L.C., M.S., E.D.R., C.C., L.P.). The remaining 137 articles were then screened by the full text.

After reviewing the remaining 137 full texts, 101 articles were excluded for the following reasons: 72 did not have a clear framework, 28 were not experimental studies, and, for 1, we could not find the complete full text.

In the end, we included only those 33 articles that met the inclusion criteria (Figure 1).

The quality was then measured using the NOS. The NOS for observational studies evaluates the quality of the study following a set of questions, in which each study can be assigned up to nine points based on three domains. The first domain, "SELECTION" (4 points), considers the selection of the study groups, the sample size, information about the responders, and whether there was a clear ascertainment of the risk factor. The second domain, "COMPARABILITY" (2 points), consists of the comparability of the different outcome groups and if the confounding factors are controlled. The last domain, "OUTCOME" (3 points), studies whether the ascertainment of the exposure and the outcome are clearly assessed, or whether the statistical test, if used, is appropriate or not.

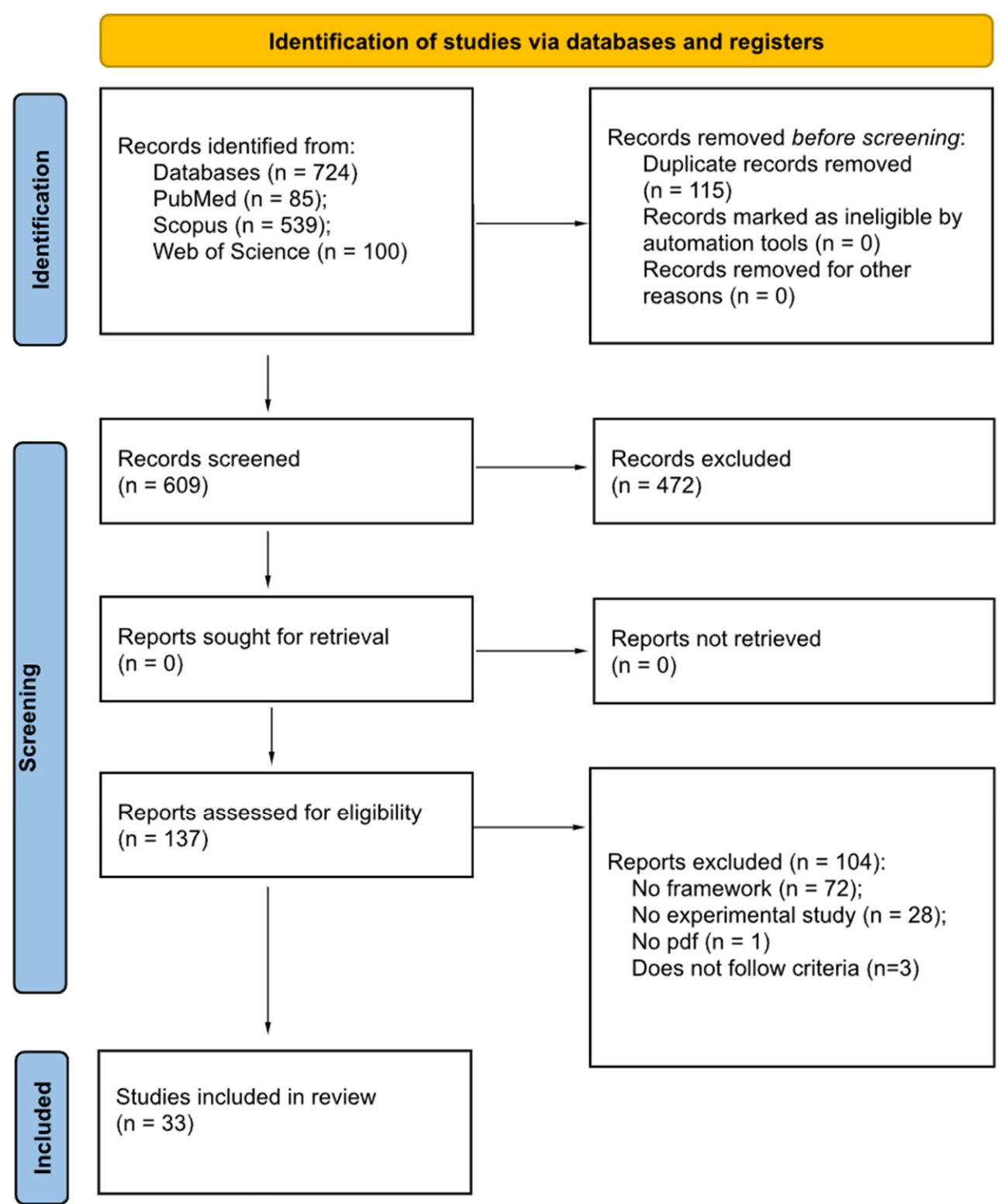

**Figure 1.** PRISMA flowchart for search strategy.

The points were then added up and the quality was "Good" if the final sum was higher than 7, "Fair" when the points assigned were between 5 and 7, or "Poor" if the final result was lower than 5.

The findings from each of those studies included are summarised in Table 1.

**Table 1.** Included studies and their characteristics.

| Author, Year, Country | Case | Methods | Parameters | Conclusion | Quality |
|---|---|---|---|---|---|
| Zeiss, C. et al., 1993, Canada [24] | Municipal waste landfill | Models for sensitivity analysis of engineering design and site selection: emission–transport models and measurements (personal sampling or ambient) and photographic evidence. | - Nuisances:<br>- Odours<br>- Noise<br>- View | Useful screening methods to predict municipal waste facility nuisance impacts. | Fair |
| Akland, G.G. et al., 1997, USA [25] | Environmental health in the lower Rio Grande Valley of Texas | Preliminary data about the levels, sources, and pathways of actual human exposure in the Valley. Samples of indoor and outdoor air, house dust, soil, food, drinking water, urine, blood, and breath were collected and analysed for several compound classes. The first sampling period was chosen to characterise the spring; the second sampling period was chosen to characterise the summer. | Questionnaires were used in this study to capture differences in lifestyle that might lead to differences in exposure to environmental contaminants.<br>Parameters analysed:<br>air—indoor; air—outdoor; house dust; water; food; blood; urine; vocs; metals; pesticides; PAHS. | This project has the potential to set a new model for environmental health research which integrates public health concerns, exposure reduction, illness prevention, and regulatory activities of many agencies. | Good |
| Fleeman, N. et al., 2000, UK [26] | The Merseyside Integrated Transport Strategy (MerITS). | HIA;<br>Identifies potential health impacts using a socioenvironmental model of health. | Biological factors (genetic; sex; age); lifestyle (diet; physical activity; recreation; means of transport; risk-taking behaviour; substance use);<br>social and economic environment (employment; culture; peer pressures; social exclusion; discrimination; community and spiritual participation);<br>physical environment (housing conditions; working conditions; water quality; air quality; noise; public safety and security);<br>access to services (education; health ± primary and secondary services; social services; housing services; transport; leisure; police; voluntary services);<br>public policy (local policies/priorities; economic; social; environmental; health; national policies/priorities; economic; social; environmental; health). | This health impact assessment identified the key health impacts of a strategy on a population and made recommendations to maximise potential positive and minimise the effects of negative impacts. | Good |
| Bonano, E. J. et al., 2000, USA [27] | Contaminated site | Integration impact assessment techniques and decision theory components in a framework that emphasises and incorporates input from stakeholders, leading to defensible decisions regarding environmental restoration and waste management. | Six categories: human health and safety, environmental protection, life cycle cost, socioeconomics, cultural, archaeological, and historical resources, and programmatic assumptions. | The integrated risk assessment–decision is a methodology for environmental management. | Good |

**Table 1.** *Cont.*

| Author, Year, Country | Case | Methods | Parameters | Conclusion | Quality |
|---|---|---|---|---|---|
| Poulsen, T.G. et al., 2002, Denmark [28] | Sludge management system | VAS; [from The italian] strategic environmental assessment. | Depletion of nonrenewable resources such as coal oil and phosphorus; contribution to global warming; acidification of lakes and forests; eutrophication of water bodies; depletion of the ozone layer; photochemical aerosol production (smog); effects on human health (pathogens, pollutants); ecotoxic effects (pollutants); land use (treatment facilities, landfills); soil quality and plant growth (sludge as soil amendment). | Strategic environmental assessment is a feasibility and utility element in decision-support systems. | Good |
| Biwer, A. et al., 2004, Germany [29] | A comparison of chemical and biotechnological production of 6-aminopenicillanic acid, a comparison of two process alternatives in the enzymatic production of α-cyclodextrin and the development of a new process for the fermentative production of pyruvate. | Modelling and simulation. | - Impact categories: raw material availability; complexity of the synthesis; critical material used; thermal risks; acute toxicity; chronic toxicity; endocrine disruption potential; global warming potential; ozone depletion potential; acidification potential; photochemical ozone creation potential; odour; eutrophication potential; organic carbon pollution potential.<br>- Impact groups: resources; grey input; component risk; organism; air; water/soil.<br>- Environmental factors: input and output components. | Method used to identify the environmental hot spots of a process and enables the comparison of process alternatives in early phases of process development. | Good |
| Monjezi, M. et al., 2008, Iran [30] | Open-pit mining and mineral processing plants | EIA, Folchi method | - Characterising the preexisting environmental context in terms of geology, geotechnics, hydrology, weather, economy, etc.<br>- Identifying the impacting factors (alteration of area's potential resources; exposition; visibility of the pit; interference with surface water; interference with underground water; increase in vehicular traffic; atmospheric release of gas and dust; fly rock; noise; ground vibration; and employment of local workforce).<br>- Defining the possible ranges for the magnitude of the variation.<br>- Singling out the environmental components whose preexisting condition could be modified as a result of mining.<br>- Correlating each impacting factor and each environmental component.<br>- Estimating the specific magnitude for each impacting factor, using the already-defined ranges.<br>- Calculating the weighted sum of the environmental impact on each environmental component. | This method provides the possibility of fair, repeatable comparisons of environmental assessments. | Good |

**Table 1.** *Cont.*

| Author, Year, Country | Case | Methods | Parameters | Conclusion | Quality |
|---|---|---|---|---|---|
| Ghasemian, M. et al., 2011, Iran [31] | Industrial estate development planning | EIA; GIS and matrix methods | GIS assessment method: identifying effective factors in environmental degradation (climate, geology, hydrology data, and different types of pollutants, land use, and ecological data). | This method is an environmental management tool before determining a plan application. | Good |
| Cordioli M. et al., 2012, Italy [32] | Waste incineration | Health risk assessment; simulation | - Study the plant and the district.<br>- Model the atmospheric dispersion of pollutants<br>- Risk estimated for the food produced in the area and for the inhalation of the emission's products.<br>- Health risk is mostly sensitive to the parameters defining the timing of exposure, such as the exposure frequency, the averaging time for carcinogenic effects, and the emission duration. Other influential parameters are the toxicological reference value and biotransfer factors between different compartments and parameters related to the food consumption for both humans and animals. Finally, parameters that determine the initial conversion from atmospheric deposition values to soil concentrations, such as the soil bulk density and soil mixing depth, play an important role. | The health risk caused by waste incineration emissions is sensitive to assumptions about the typical diet of the resident population, and the geographical origins of food production. | Good |
| McKenzie, L. M. et al., 2012, USA [33] | Unconventional natural gas development | HIA: Use of standard United States Environmental Protection Agency (EPA) methodology to estimate noncancer HIs and excess lifetime cancer risks for exposures to hydrocarbons. | Air toxics data (56 hydrocarbons). Risks were estimated for two populations: (1) residents > ½ mile from wells; and (2) residents ≤ ½ mile from wells. Exposure scenarios were developed for chronic noncancer HIs and cancer risks for 30 years. | Comparison of risks between residents based on proximity to wells illustrates how the risk assessment processes can be used to support the HIA process. | Good |
| Ni, H. et al., 2014, China [34] | Accidental pollution in the chemical industry | Environmental numeric simulation models, model integration methods, and modern information technology integrated into the WEB Geographic Information System (WEBGIS) platform. | Leakage: atmospheric pollutant diffusion simulation. Parameters: leakage type; height of source; molecular weight of source air; concentration of average time; heat capacity in the fixed pressure; largest distance calculation; boiling point temperature; height of the concentration calculated; evaporation heat consumption; initial water content; liquid heat capacity; source temperature; liquid density of source gas; release intensity of source; saturated vapour pressure parameter; source area; unsaturated vapour pressure parameter; release duration; environmental height—stations; relative position x of source; environment temperature; relative position y of source; relative humidity; surface roughness; stability; wind direction; environmental wind. | This method could provide effective support for deciding emergency responses of acute chemical accidents and project a safe implant. | Good |

**Table 1.** *Cont.*

| Author, Year, Country | Case | Methods | Parameters | Conclusion | Quality |
|---|---|---|---|---|---|
| Lam, S. et al., 2014, Canada [35] | Grey Bruce Health Unit | Aimed for comprehensiveness in data compilation, including the following: standard media categories (e.g., air, water, land); and ecological indicators (e.g., vectors, forests, wetlands). Data sources included both primary (collected by an organisation) and secondary (assembled by others). | Driving forces (economic, social, political, technologic, institutional), pressures (ecosystem depletion, waste release), state (degraded ecosystems, pollution), exposures (pollutants, infectious agents), effects (disease, mortality), action. | The results suggest that routinely collected environmental and health data can be structured into the framework, though challenges arose due to gaps in data availability, particularly for social and gender analyses. | Good |
| Palmieri, N. et al., 2014, Italy [36] | Rapeseed production | Life Cycle Assessment | Impact of machinery, fertilisers, seeds, herbicides/pesticides, technical characteristics of tractors and agricultural equipment, diesel consumption. Air, water, and soil emissions caused by nitrogen fertilisers in the soil and by tractors. | The environmental assessment carried out in the paper identified the impacts of units, processes, and cultivation practices that are more responsible for some environmental issues. Moreover, it has shown how these results could be influenced by the yield per hectare. The integration of environmental analysis with economic considerations allows for some conclusions to be drawn. | Good |
| Hu, H. et al. 2015, China [37] | Waste Incineration Plants | EIA; multi-criteria decision analysis | Distance from surface water, land use suitability, wetlands, distance from water sources, distance from residential areas, traffic, distance from flight paths, distance from infrastructure and power lines, rainfall, air pollution index, distance from railway, odour, floodplains, distance from natural springs, distance from irrigational canals, distance from highway, distance from forest lands, distance from tourism areas, ecological impacts, distance from leisure areas, distance from archaeological sites, distance from burial yards, distance from other special areas, noise, dust. | The EIA model is important to protect the environment and health of the residents living around these plants. Public participation can play a role in the supervision. | Good |

Table 1. *Cont.*

| Author, Year, Country | Case | Methods | Parameters | Conclusion | Quality |
|---|---|---|---|---|---|
| Baum, F.E. et al., 2016, Australia [38] | Transnational corporations | Based on ex post assessment and follow the standard HIA steps of screening, scoping, identification, assessment, decision making, and recommendations. | Workforce and working conditions (e.g., description, occupational health systems, remuneration of workers in relation to cost of living indexes, extent of unionisation, quality of provision of healthcare, and impact on social determinants of health such as housing). Social conditions (e.g., impact of TNC goods on locally produced goods and services and net employment levels, impact of operation on local living conditions, the value of corporate social responsibility initiatives, social dynamics created by TNC operations including impact of fly-in–fly-out workers, impacts on social, cultural, and spiritual life, and the impact of migrant labour in mines affecting sexual practices). Environment (e.g., impact on natural systems in ways that affect health or health risk, including air/water quality, exposure to pollutants, land clearing, energy consumption, water, waste disposal). Consumption patterns (e.g., impact of quality and consumption of TNC goods on health, national marketing practices). Economically mediated impact on health (e.g., impact on TNC operations on overall economic conditions including tax revenues, reliance and vulnerability of national economy on TNC, economic and health impacts on local businesses/farmers). | The results would be available for use by civil society advocates, corporations who wish to lessen the adverse health impact of their operations and by governments who would be able to assess different regulatory frameworks according to their ability to reduce adverse health and equity impacts and/or enhance health benefits of TNC operations. | Good |
| Kim, T. H. et al., 2016, Republic of Korea [39] | Concrete production process | Life Cycle Assessment | Impact assessment is divided into four steps: (1) classification; (2) characterization, (3) normalisation; and (4) weighting in which relative importance among the impact categories is determined as global warming (GWP), acidification (AP), eutrophication (EP), abiotic depletion (ADP), ozone depletion (ODP), and photochemical oxidant creation (POCP). | These case analysis results allow the assumption that single-category environmental impact assessment cannot yield any reliable assessment results regarding the eco-sustainability of concrete, which requires multi-category assessment. | Good |

**Table 1.** *Cont.*

| Author, Year, Country | Case | Methods | Parameters | Conclusion | Quality |
|---|---|---|---|---|---|
| Yost, E.E. et al., 2017, USA [40] | Hydraulic fracturing to impact drinking water resources | Multi-criteria decision analysis (MCDA) framework. A toxicity score, an occurrence score, and a physicochemical properties score. | - Toxicity: (1) a noncancer MCDA, in which the toxicity score is calculated using RfVs; (2) a cancer MCDA, in which the toxicity score is calculated using OSFs.<br>- Occurrence: the frequency at which each chemical was reportedly used in hydraulic fracturing fluids.<br>- Physicochemical Properties: inherent physicochemical properties which affect the likelihood that a chemical will be transported in water. (simulated)<br>- Mobility score: based upon three physicochemical properties that describe chemical solvency in water: the octanol/water partition coefficient (KOW), the soil adsorption coefficient (KOC), and aqueous solubility.<br>- Volatility Score: Henry's law constant.<br>- Persistence Score: based on estimated half-life in water. Total physicochemical properties score—For each chemical, the mobility score, volatility score, and persistence score (each on a scale of 1 to 4) were summed to calculate a total physicochemical properties score. Within each MCDA (noncancer or cancer), the three criteria scores (toxicity, occurrence, physicochemical properties) were each standardised to the dataset by scaling to the highest and lowest respective score within the given subset of chemicals. Comparison with field data. Knowledge of site-specific variables. | This approach is a preliminary analysis, useful to explore potential public health. | Good |
| Oduro-Appiah, K. et al., 2017, Ghana [41] | The municipal solid waste management system | Integrated solid waste management (ISWM) | The quantitative indicators, respectively, measure the following: (1) the percentage of households with access to a reliable collection service; (2) the proportion of the total MSW generated that is captured by the management system. The qualitative indicator determines the quality of collection based on six multi-attribute composite criteria. | The analysis suggests that waste and recycling would improve through greater provider inclusivity, especially the recognition and integration of the informal sector, and interventions that respond to user needs for more inclusive decision making. | Good |

**Table 1.** *Cont.*

| Author, Year, Country | Case | Methods | Parameters | Conclusion | Quality |
|---|---|---|---|---|---|
| Sajjadi, S.A. et al., 2017, Iran [42] | Municipal waste landfill | EIA, Leopold matrix | Evaluation of four environments (physical, biological, economic, and social) for each activity (current landfill, compost plant, recycling plant, incineration, sanitary landfill). After scoring the matrix, the results were concluded in Excel software. In all options studied in the construction and operation phases, most negative effects on the physical, biological, and socioeconomic environments were related to noise pollution, air quality, land ecosystems, income and costs, and an increase in real estate prices, respectively; in the case of cultural environment, most negative effects were related to landscapes and social acceptance. | This method provides a useful score for the decision-making phase. | Good |
| Lohse, C., 2017, Germany [43] | Hydrogeothermal energy generation | Life Cycle Assessment | Determine the interaction with other environmental and conservation objectives; identify the short-, medium- or long-term harmful effects on human health, the environment, and cultural heritage, which are substantially induced by use of natural resources as well as material or energy releases; introduce and establish environmentally friendly optimised technologies, products and concepts. | The Life Cycle Assessment shows that environmental impacts from geothermal binary plants for power and heat supply are strongly influenced by the geological site preconditions. | Fair |
| Mueller, N. et al., 2017, Spain [44] | Urban and transport planning | Associations between exposures and morbidities and calculated population attributable fractions to estimate the number of attributable cases | Ischemic heart disease, hypertension, stroke, type 2 diabetes mellitus, colon cancer, breast cancer, dementia, depression, traffic incidents with injuries, respiratory hospital admissions, fecundity, preterm birth, low birth weight cases attributable to non-compliance of international exposure recommendations of physical activity, air pollution, noise, heat, and access to green spaces. | (1) The reduction in motor traffic together with the promotion of active transport and (2) the provision of green infrastructure would result in a considerable BD avoided and substantial savings to the public healthcare system, as these measures can provide mitigation of noise, air pollution, and heat as well as opportunities for promotion. | Good |

**Table 1.** *Cont.*

| Author, Year, Country | Case | Methods | Parameters | Conclusion | Quality |
|---|---|---|---|---|---|
| Chen, L. et al., 2019, China [45] | Impact of land use planning on the atmospheric environment | A methodology combining the land-use-based emission inventories of airborne pollutants and the long-term air pollution multi-source dispersion (LAPMD) model. | An emission inventory of airborne pollutants can provide spatial source intensity for dispersion assessment. Emissions from individual land use types in the reference and target years can be estimated using their respective inventories. By means of the LAPMD model, spatial variability of airborne pollutants in the reference and target years can be quantified, and the LUP impact on the atmospheric environment can be assessed. | Land-use-based emission inventorying is a more economical way to assess the overall pollutant emissions compared with the industry-based method, and the LAPMD model can map the spatial variability of airborne pollutant concentrations that directly reflects how the implementation of the land-use planning (LUP) scheme impacts on the atmosphere; (2) the environmental friendliness of the LUP scheme can be assessed by an overlay analysis based on the pollution concentration maps and land-use planning maps; (3) decreases in the emissions of $SO_2$ and PM10 within Lianyungang indicate the overall positive impact of land-use planning implementation, while increases in these emissions from certain land-use types (i.e., urban residential and transportation lands) suggest the aggravation of airborne pollutants from these land parcels; and (4) the city centre, where most urban population resides, and areas around key plots would be affected by high pollution concentrations. | Good |

**Table 1.** *Cont.*

| Author, Year, Country | Case | Methods | Parameters | Conclusion | Quality |
|---|---|---|---|---|---|
| Masum, S.A. et al., 2020, UK [46] | Direct discharge of untreated tannery waste in the environment | Temporal and spatial distribution of four heavy metals: chromium (Cr), lead (Pb), cadmium (Cd), and arsenic (As) have been modelled using a numerical model, namely COMPASS, which studies coupled fluid flows, reaction and deformation processes in subsurface porous media. | The model investigates heat, liquid, moisture, gas, and chemical flows, microbial, geochemical, and biogeochemical reaction processes, and mechanical deformation under a coupled framework. | This information is important for a comprehensive environmental impact assessment. | Fair |
| Li, Y. et al., 2020, China [47] | Coal-to-gas conversion | Life cycle assessment and life cycle cost methods | Environmental Impacts: Climate change, freshwater eutrophication, human toxicity, particulate matter formation, freshwater ecotoxicity, and marine ecotoxicity. Life cycle inventory: Electricity for cooking and other electricity consumption, maize straw, bulk coal, natural gas for cooking and for heating, infrastructure construction. | Switching from coal to gas reduces environmental impacts significantly and in particular will assist with the massive air pollution problem that is concerning tens of millions of people. This in turn will have significant health benefits by improving both indoor and outdoor air quality. | Good |
| Haigh, F. et al., 2020, Australia [48] | New greenfield airport | HIA, VIS; the data were collected through workshops with affected communities; an online survey; a review of the peer-reviewed and grey literature; and local and state-level socio-demographic and health data. | 1. How are communities currently receiving information about airport planning? 2. How are communities currently engaged in airport planning processes? 3. What is the current status of wellbeing in potentially affected communities? 4. How do communities perceive the information they receive and the way they are engaged in planning decision making and how is it affecting them? | This method shows the need for community engagement efforts to ensure that airports can promote health and wellbeing both during development and throughout operations. | Good |
| Arani, M.H. et al., 2021, Iran [49] | Steel industry development plan | EIA; combined method involving Leopold matrix and RIAM | Evaluation of the impact: physiochemical, biological, economic, social, and cultural aspects and the pollutants emitted from hot rolling processes into the ambient air of the region. The data for investigation of environmental factors, map of surface and groundwater resources, weather and natural resources were collected from various public organisations. The data were analysed using Leopold and RIAM matrices in RIAM and Microsoft Excel software. Environmental impacts of the development plan were assessed by combining two methods of Leopold and Pastakia matrices. | Decisions made because of the scores obtained regarding positive impacts and negative impacts. | Good |

**Table 1.** *Cont.*

| Author, Year, Country | Case | Methods | Parameters | Conclusion | Quality |
|---|---|---|---|---|---|
| Dawoudian, J. et al., 2021, Iran [50] | Cement industries | Mathematical matrix method | Description of the project and environmental characteristics. Identification and prediction of the effect. Valuation of the significance of the project's description and the characteristics of the environment. Environmental factors: air pollution and micro-climate; water pollution; sound pollution; soil pollution; biodiversity; socioeconomic and cultural environment. | Environmental Impact Assessment is an indispensable tool for proper implementation of major projects. Thus, developers should provide methods to eliminate, reduce, or control possible adverse environmental effects and provide the possibility of renewal, restoration, and compensation of damage to the environment. | Good |
| Sarigiannis, D.A. et al., 2021, Greece [51] | Waste management | Life cycle assessment. Collection of the data on all environmental interventions in the unit processes (inventory phase), conversion of inventory data into environmental effects (impact assessment phase), and interpretation of the results in relation to the objectives of the study. The health impacts considered long-term mortality and morbidity including carcinogenicity, premature mortality, decreased birth rate, and increased incidence of congenital anomalies in neonates, considering the excess risk over forty years. | (a) Global warming potential expressed in terms of $CO_2$ equivalent. (b) Acidification potential expressed in terms of $SO_2$ equivalent. (c) Tropospheric ozone precursor potential. (d) Environmental emissions. | Life cycle analysis produces different conclusions than a simple environmental impact assessment based only on estimated or measured emissions. | Good |
| Kim, M.K. et al., 2021, Republic of Korea [52] | Railroad Development Areas | Independence analysis and logistic regression analysis. | Biodiversity class, ecosystem type, vegetation conservation class, tree age class, ecological naturalness, presence of river ecosystems, and fragmented patch size. | Based on the regression model, a probability map of environmentally favourable areas and an environmental quality evaluation map were constructed. The results of this study can be applied to railway development project sites and may help to identify the best sites for the development of an environmentally friendly railway system. | Good |

**Table 1.** *Cont.*

| Author, Year, Country | Case | Methods | Parameters | Conclusion | Quality |
|---|---|---|---|---|---|
| Ponghiran, W. et al., 2021, Thailand [53] | Gold extraction processes for discarded computer RAM | Life Cycle Assessment | Terrestrial ecotoxicity and human carcinogenic toxicity | The comparison between the two leaching processes without waste management demonstrated that the cyanide-based solution provided 8.5 times lower in terrestrial ecotoxicity and 6.4 times lower in human carcinogenic toxicity than aqua regia due to the lower overall chemical consumption. | Good |
| Tianliang, W. et al., 2023, Iran [54] | Coal mine | EIA; these methods include checklists, matrices, and networks. | Physical/chemical (noise pollution, air pollution, water pollution, soil pollution, etc.); biological/ecological (plants, animals, habitats, etc.); sociological/cultural (population, migration, traffic, health and education indicators, welfare, etc.); economic/operational (employment, income from coal, land prices, etc.). | The method proved to be transparent because it indicates clearly where the scores are coming from by indicating the environmental components that were affected. | Good |
| Armanuos, A. M. et al., 2023, Egypt [55] | Landfill | GIS and a multi-criteria decision making (analytical hierarchy procedure, ratio scale weighting, straight rank sum, and Boolean method). | Natural criteria (groundwater, surface water, soils, elevation, slope, and land use) and artificial criteria (roads, railways, urban areas, villages, and power lines). | The Boolean method is limited, while the three methods (analytical hierarchy procedure, ratio scale weighting, straight rank sum) are similar. The best result is obtained with the integration of the three methods and is founded upon the selected criteria and the availability of data. | Good |
| Tao, M. et al., 2023, China [56] | Coal Mine | Life Cycle Assessment | Global warming (GW), stratospheric ozone depletion (SOD), ionising radiation (IR), ozone formation human health (OF-HH), fine particulate matter formation (FPMF), ozone formation terrestrial ecosystems (OF-TE), terrestrial acidification (TA), freshwater eutrophication (FEu), marine eutrophication (MEu), terrestrial ecotoxicity (TEu), freshwater ecotoxicity (FEc), marine ecotoxicity (MEc), human carcinogenic toxicity (HCT), human noncarcinogenic toxicity (HnCT), land use (LU), mineral resource scarcity (MRS), fossil resource scarcity (FRS), and water consumption (WC). | This method shows that the greater the proportion of the total environmental impact in the production stage. | Good |

The articles included were published between the years 1993 [24] and 2023 [54–56]. Studies were conducted in Canada [24,35], the USA [25,27,33,40], the UK [26,46], Denmark [28], Germany [29,43], Iran [30,31,42,49,50,54], Italy [32,36], China [34,37,45,47,56], Australia [38,48], Republic of Korea [39,52], Ghana [41], Spain [44], Greece [51], Thailand [53], and Egypt [55].

The included studies covered a variety of contexts, considering from landfill plans [24,28,32,37,41,42,46,51,55], manufacturing factories [29,31,39,49,50,53], mobility actions [26,38,44,48,52], energy productions [30,33,36,40,43,47,54,56], and the environmental health of an area [25,27,34,35,45].

Each article describes a different method for the impact assessment. The HIA method is one of those used by the authors [26,27,33,38,48]. Another method is the EIA [30,31,37,42,49,54]. Several authors have assessed the possible impact of the process throughout the entire procedure, using the Life Cycle Assessment method [36,39,41,43,47, 51,53,56] and a Life Cycle Cost method [47].

Another technique that was also equally represented is one that involves the use of mathematical simulators and a multi-criteria decision analysis [24–27,29–35,37,40,42,44–46,49,50,52,54,55]. This method can also be combined with the information extracted from the Geographical Information System (GIS) [31,34,55]. Only a study carries out an SEA [28] and a Specific Health Impact Assessment [48].

Determinants of health are assessed as parameters with a focus on human health and safety [26–29,32,33,35,40,42–44,47–49,54,56], environmental impact [24–43,45–47,49–52,54–56], the contribution to global warming [28,29,32,39,51,56], eco-toxic effects [28–32,35,47,50,52,53,56], human toxicity [28,29,35,40,47,53,56], life cycle cost [27,35,41,47], and socioeconomical [26,27,35,38,42,49,50,54], cultural [26,27,38,42,43,49,50,54], archaeological and historical resources [27], and programmatic assumptions [27,35,41,47]. Biomonitoring of blood and urine samples can also be used to assess human health [25]. All studies show how an approach which takes into account many parameters can help analyse the impact of a new facility in a much more comprehensive and thorough manner.

Regarding the quality assessment, only 3 studies were considered "Fair", while 30 were considered "Good".

## 4. Discussion

The results of this review showed the importance of using EIA tools in order to minimise the effect of anthropogenic activities. The advantage of using these tools in the preliminary stages of study (subjectivity verification, screening, etc.) is to enable decision makers to choose between various alternatives with respect to the future consequences of the choices that are to be implemented [57].

As highlighted in the literature, it is essential, in the context of Impact Assessment Processes, whether environment- or health-related, to evaluate both direct impact factors: for instance, those concerning emissions (even during the processing phase) and indirect impact factors, such as those related to the socioeconomic status of the population. In this regard, it is useful to remember that there is a growing interest in the literature in defining the so-called "double exposure" [58], i.e., environmental and social factors. Moreover, within the studies on the social determinants of health, these are characterised as being, on the one hand, directly correlated with the health status of the population, but also as proxies of the effect of environmental exposures. Given the same level of exposure, a socioeconomically disadvantaged population will have worse health outcomes than one with a higher socioeconomic status [59]. At this point—and this is one of the main limitations of the HIA methodology, at least for the literature included in the study—the field becomes less defined; it shifts from a deterministic concept of health to one related to vulnerability. This is a debated theme and is difficult to define with the tools of classical epidemiology. In this sense, it is therefore useful to specify how, in all phases of the impact assessment processes, interdisciplinarity plays a fundamental role in understanding health dynamics in light of the population's life trajectories [60]. This also addresses more

complex themes, where causal links blur in qualitative assessment, and where the work of health professionals becomes indispensable and cannot be replaced only by "remote" evaluations [61].

In this context, it is important to distinguish the roles between social determinants of health, which, as described above, have an epidemiological value as effect modifiers, from incident factors, such as pollution. The former are evaluated, as it emerges from the review, in terms of "indirect factors"; therefore, they only marginally fall within the strictly environmental assessment; the latter are, in all the studies included, the true focus of the procedure. The purpose of this review is solely to evaluate which elements are internationally included in EIA procedures, not to explore the causal links and possible mediations between incident factors and population health. This latter area is the subject of more quantitative research that requires robust statistical tools and extensive data sources.

Additional elements to consider, particularly with regard to the assessments in the context of procedures related to plans, concern various factors and how they interact, starting from a regional or metropolitan scale level of planning. For example, on the theme of mobility, urban architectural factors affecting active cycling and walking mobility, and how these influence the propensity to use one mode of transportation over another [62].

In this sense, it would be appropriate to delve deeper, within the context of impact assessment procedures, into the theme of environmental, social, and therefore health co-benefits of interventions related to mobility or new urban developments [63], and determine how these fit into contexts; this is particularly the case in light of the available evidence on the relationship. For example, the relationship between green areas and health [64], as it is mediated by socioeconomic factors like the real estate market [65].

Returning instead to the procedure itself, the decision can be facilitated if the initial impact assessment is as wide as possible and examines as many of the analysable parameters as feasible. The elements investigated should be as varied as possible. This is in order to analyse all aspects of the life of the community involved, and also to reach a more equitable and sustainable decision. When evaluating a newly developed anthropogenic activity, it is advisable to consider the surrounding environment, including GIS studies, in order to define a location that has the lowest impact on the social, territorial, and environmental context. It is also important to consider the historical, archaeological, and cultural context of the area, as well as the impact that the facility may have on the current economic reality and infrastructure. In every production process, even the simple transport of raw materials, molecules are generated at every stage of the cycle, whose effect on the environment and ecology can be minimised if they are studied, analysed, and researched. It is therefore necessary to know their chemical and physical properties, which also include stability and reactivity, toxicological and ecological information, as well as normal disposal and transportation procedures. In addition to all of this information, it would be appropriate to add that of the population, on which the new anthropogenic activity will have a real impact. For example, it is useful to know the average age, health status, lifestyles, living and working conditions, social and community networks, and the socioeconomic situation of the community, and to make a forecast of how these factors could change, enhancing the use of instruments like longitudinal studies [66].

This prospective study, essential in order to allow decision makers to choose alternatives, should also be followed by an epidemiological, environmental, socioeconomic, and cultural assessment some duration of time after the construction of a plant, in order to be able to intervene and, possibly, implement corrective procedures.

Finally, it is useful to specify how the regulatory framework varies from country to country, and how this is decisive in defining the perimeter within which the authority responsible for prevention (or otherwise expressing opinions within the context of impact assessment processes) can or cannot act. In this sense, it would be useful to be able to define—and this could concern further future studies—the relationship between existing regulations, the competencies assigned to authorities, and the possibility of influencing the

proceedings in order to balance the needs of production and economic development with those of environmental sustainability and public health.

This review has some limitations. First, we could not perform a meta-analysis due to the high heterogeneity of the studies examined. This is due the characteristics of the anthropogenic activities evaluated and, in particular, the variety of methodologies and the parameters employed. Another limitation is the legislative variability between the different countries that carried out the studies, which does not allow a direct comparison between these studies. This highlights the importance of integrating methodologies as much as possible in order to expand the number of parameters used and to better define the needs for global health.

## 5. Conclusions

The EIA is a regulated tool of fundamental importance to Public Health. The more parameters are analysed, the clearer the choice with the least social and environmental impact is. The inclusion of a HIA, which considers all determinants of health, and not just environmental determinants, in the process of development, is of major importance. Putting community wellbeing before the economic aspects at the planning stage can minimise the impact of the new anthropogenic activity.

**Author Contributions:** Conceptualization, L.C., M.S. and L.P.; methodology, L.C.; software, L.C., M.S. and L.P.; validation, L.C., M.S., E.D.R., C.C. and L.P.; formal analysis, L.C.; investigation, L.C., M.S., E.D.R., C.C. and L.P.; resources, L.C. and M.S.; data curation, L.C.; writing—original draft preparation, L.C. and M.S.; writing—review and editing, L.C., M.S., E.D.R., C.C. and L.P.; visualization, L.C., M.S., E.D.R., C.C. and L.P.; supervision, E.D.R. and L.P. All authors have read and agreed to the published version of the manuscript.

**Funding:** This research received no external funding.

**Data Availability Statement:** All data presented in this study are available on request from the corresponding author.

**Conflicts of Interest:** The authors declare no conflicts of interest.

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
