# Peer review of "Evaluating the Environmental Impact of Anthropogenic Activities on Human Health: A Systematic Review"

_urbansci, doi:10.3390/urbansci8020049_

Round 1
Reviewer 1 Report
Comments and Suggestions for Authors
I appreciate the authors for their scientific contribution. Although this manuscript provides a comprehensive and detailed summary of the environmental impact of anthropogenic activities on human health, the authors still need to work on some improvement. The following inputs and comments could be considered.
ABSTRACT
Please clearly the types and characteristics of various assessment systems (The results section also has the same issue, please elaborate them in the results section).
Please indicate this review's practical significance and provide suggestions for future research.
INTRODUCTION
Lines 38-39 - "to ensure that all different aspects of the long term effects" Does this assessment (SEA) cover "all" aspects? Is there evidence to support this claim? Or could you use a less definitive way of expressing it?
Lines 56-59 - Some references should be added to increase persuasiveness. Previous studies have shown the impact of living environment on human health, e.g.: "Underestimated or overestimated? Dynamic assessment of hourly PM2.5 exposure in the metropolitan area based on heatmap and micro-air monitoring stations" (https://www.sciencedirect.com/science/article/abs/pii/S0048969721013516 ); there are also studies confirming the environmental injustices faced by low-income people, which can be supported by citing "A study on air pollution exposure of ‘first and last mile’ urban commuters under space-behavior dual verification based on big data, land-use regression model and space syntax" (https://www.sciencedirect.com/science/article/abs/pii/S0959652623024022).
Lines 63-68 - The SEA incorporates the VIA? The sequencing and relationship between them can be confusing. It is recommended to clarify the logic of each assessment in the introduction.
Lines 70-71 - The sentence - "This Decree tends to have a very specific flaw as it focuses on the ambient and, therefore, the health impact is not easily measured" and the first sentence of the following paragraph convey similar meanings, and what is the relationship between them?
RESULTS
Line 136 - “605” should be changed to “609”? In combination with Figure 1, it should be elaborated in detail how 137 articles were selected from the 609 references, rather than simplifying it as "records excluded".
Table 1 - What are the evaluation criteria for "Quality"? Please elaborate.
Table 1 - The table should be labeled with numbers for easy reference and correlation with the summary below.
DISCUSSION
The discussion section now is focused on listing the specific contents of various assessment systems, lacking quantitative comparisons. Please classify existing assessment systems based on literature, summarize the effects of assessment systems, discuss the advantages and disadvantages of existing assessment, and present results in a table or figure. Suggestions for improvement based on quantitative results also need be provided.
Comments on the Quality of English LanguageIn general, there are some English grammar issues, it's better to revise the article once again or send it to professional English proofreader for a smoother finish.
ABSTRACT
Line 14 - “industrial and production”— Adjectives and nouns placed side by side?
INTRODUCTION
Line 31 - “and, also, ”Is it a little colloquial?
Lines 42-43 - " Between 1973 and 1974 Australia, Canada and New Zealand……” Please check the preposition collocations.
Author Response
Dear Sir/Madam,
On behalf of the authors, I would like to thank you very much for taking the time to review this manuscript. Please find enclosed our reply.
Yours sincerely,
MS

Reviewer 2 Report
Comments and Suggestions for Authors
Thank you for your preparation of the manuscript. I believe you have done a thorough review of the literature.
My only suggestion is to expand your results sections further on the human health indicators in the study. You mention the social determinants of health, but this is very broad. The readership could benefit from understanding what determinants were used in the study to capture the impact on human health especially since you make the argument that we need to incorporate it better into how we do assessments.
Author Response

(The authors gave the same response as above.)
